



# Nuclear magnetic resonance free ligand conformations and atomic resolution dynamics

**Amber Y. S. Balazs**[1], **Nichola L. Davies**[2], **David Longmire**[2], **Martin J. Packer**[2], and **Elisabetta Chiarparin**[2]

[1]Chemistry, Oncology R&D, AstraZeneca, Waltham, Massachusetts 02451, United States
[2]Chemistry, Oncology R&D, AstraZeneca, Cambridge CB4 0QA, United Kingdom

**Correspondence:** Amber Y. S. Balazs (amber.balazs@astrazeneca.com)

**Abstract.** TS1 Knowledge of free ligand conformational preferences (energy minima) and conformational dynamics (rotational energy barriers) of small molecules in solution can guide drug design hypotheses and help rank ideas to bias syntheses towards more active compounds. Visualization of conformational exchange dynamics around torsion angles, by replica exchange with solute-tempering molecular dynamics (REST-MD), gives results in agreement with high-resolution [1]H nuclear magnetic resonance (NMR) spectra and complements free ligand conformational analyses. Rotational energy barriers around individual bonds are comparable between calculated and experimental values, making the in-silico method relevant to ranking prospective design ideas in drug discovery programs, particularly across a series of analogs. Prioritizing design ideas, based on calculations and analysis of measurements across a series, efficiently guides rational discovery towards the "right molecules" for effective medicines.

## 1 Introduction

Nuclear magnetic resonance (NMR) signal line shapes inherently provide atomic-level, site-specific insights into structural dynamics. High-resolution [1]H NMR signals broaden when small molecules in solution undergo exchange dynamics on a millisecond timescale. In contrast, sharp NMR signals may indicate either a dominant pre-organized conformation or an ensemble of flexible molecules undergoing fast equilibrium exchange between rotational isomers. Comparisons between experimental and computed NMR parameter values (shifts, NOEs, $J$ couplings) can identify relative populations of conformers, such as a singular, highly populated conformation, with well-defined internuclear distances and torsion angles or an averaged solution structure, composed of multiple conformations, each at a low molar fraction of the total, resulting from low barriers to rotation around bonds. NMR analysis of molecular flexibility in solution (NAMFIS; Cicero et al., 1995) takes the approach of systematically varying percent contributions from sets of conformers generated in-silico, together with the corresponding *calcu-lated* NMR parameter values, compared against the experimental data. The sum of square differences determines the goodness-of-fit between experimental and calculated values to select a best-fit population-weighted model. The fundamental concept of filtering theoretical conformations through experimental data to derive the best fit has become well established over the decades, together with variations in details of implementation, to determine the conformational preference(s) of a small molecule in solution (Cicero et al., 1995; Nevins et al, 1999; Slabber et al., 2016; Wu et al., 2017; Balazs et al., 2019; Farès et al., 2019; Atilaw et al., 2021).

Determining the conformational profile of a free ligand in solution enhances early drug discovery programs (LaPlante et al., 2014; Blundell et al., 2016; Foloppe and Chen, 2016; Chiarparin et al., 2019). A general overview of how NMR fits into the design–make–test–analyze (DMTA) cycle is illustrated in Fig. 1. An efficacious pharmaceutical that positively impacts patients' lives starts with medicinal chemistry teams designing the right molecule. Design teams need to understand whether a molecule readily adopts its "bioactive" conformation in solution to optimize the binding on-

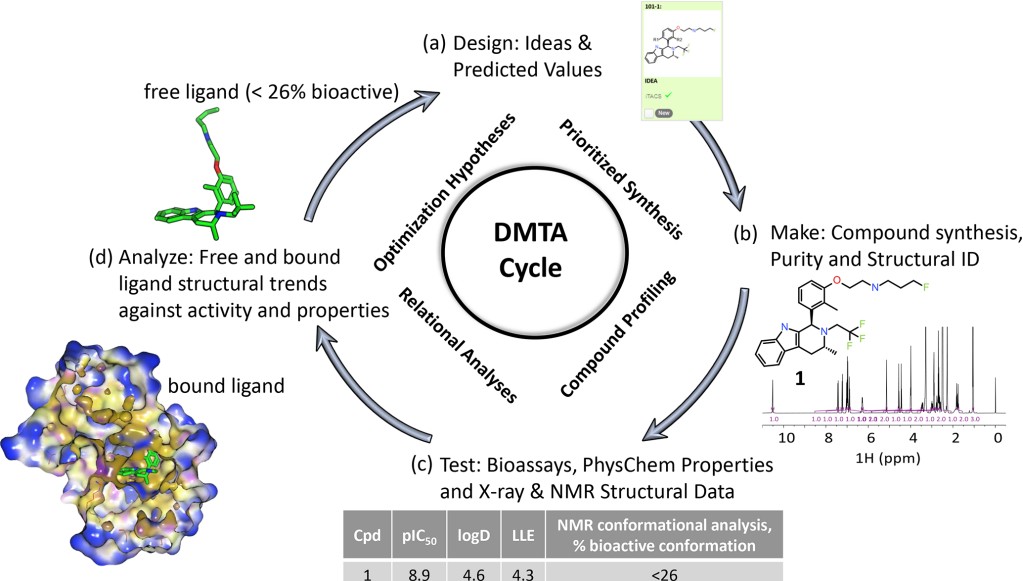

**Figure 1.** An illustrative medicinal chemistry design–make–test–analyze (DMTA) cycle for drug discovery; the example shown is taken from a recent Oncology R&D project (Scott et al., 2016, 2019, 2020). **(a)** Design: medicinal chemists design drug molecule ideas, using predicted values from computational models to prioritize which virtual molecules to synthesize. **(b)** Make: compounds are synthesized and NMR plays a key role in structural identification and analysis of compound purity. **(c)** Test: compound profiling includes bioassays to quantify activity, such as target inhibition (pIC50), and physico-chemical properties, such as the octanol–water partition coefficient (logD). X-ray structure of the bound ligand–protein complex and NMR free ligand conformations are measured, including the relative population of the bioactive bound conformation (Balazs et al., 2019). **(d)** Analyze: compound free and bound structures are analyzed against measured properties to rationalize structure activity and property relationships to derive new hypotheses for improved designs in step (a) of the cycle. Typical discovery projects comprise ∼ 1000 cycles from hit to drug candidate.

rate through reduced conformational entropy and energetic penalty paid on conformational rearrangement to the proper binding mode. In addition, pre-organization of the free ligand in solution would indicate minimized conformational strain energy in the bound molecule. If not, the challenge is to conceive of ideas to modify the structure to discover a new molecule to favor this conformation. Towards this aim of optimizing the free energy of binding, it is desirable for ligands in solution to preferentially pre-organize into the bioactive binding mode (Blundell et al., 2013; Balazs et al., 2019). Molecular rigidification strategies (Fang et al., 2014; de Sena M Pinheiro et al., 2019) increase pre-organization, and NMR conformational analysis can deconvolute and report on the molar fraction adopting the bioactive conformation. Structure-based drug design (SBDD) can be enhanced by ready access to 3D free ligand average solution conformations to complement X-ray crystallographic models of the bound ligand and protein–ligand interactions (Blundell et al., 2013; Chiarparin et al., 2019; Balazs et al., 2019). Faster design cycles require quick turnover times in analyzing solution conformations of synthesized compounds. Design cycles can be accelerated through faster computational schemes, efficient automation to obtain NMR spectral parameters, and recognition of conformational signatures from 1D NMR spectra (Balazs et al., 2019), also named "SAR by 1D NMR" (Zondlo, 2019).

Herein, we demonstrate incorporation of molecular dynamics, specifically an efficient version using replica exchange with solute tempering (REST-MD) (Liu et al., 2005; Huang et al., 2007; Wang et al., 2011), into an NMR-based semi-automatic drug discovery platform, to visualize rotational barriers around molecular bonds. Good agreement is demonstrated between REST-MD-calculated energy barriers and NMR measurements, using a small-molecule selective estrogen-receptor degrader (SERD) example from a recent Oncology R&D project (Scott et al., 2016, 2019, 2020). The theoretical and experimental data complement each other: REST-MD simplifies the interpretation of NMR conformational dynamics, while the experimental NMR results can inform calculations by defining site-specific preferred torsions of the dominant conformer and experimental conformer distributions, which may influence the initial REST-MD 3D geometry and the sampling ergodicity achieved, as reflected in the resultant histograms.

## 2 Results and discussion

### 2.1 NAMFIS plus NMR line shape analysis

The ability of NMR to provide information on conformational dynamics, in addition to giving information on conformational preference, is useful in small-molecule drug discovery. In Fig. 2a, each peak is doubled for compound 2 (1 : 1 ratio), instantly recognizable to NMR users as slow exchange of rotamers due to hindered bond rotation (measured half-life $\sim 0.5$ s). As a guide to the eye, the signal(s) for the benzylic CH proton at $\sim 5$ ppm is/are highlighted in Fig. 2. The NMR spectrum reports two dominant conformers, equally populated, for the free ligand in solution for compound 2. The bioactive conformation is one of a family of conformers (shown in green) with some flexibility around the pendant base. The alternative conformation (shown in orange) is the other, giving $\sim 50$ % of the compound locked in a non-bioactive conformation. In contrast, Fig. 2b shows that compound 3 has a single set of sharp signals due to fast exchange (corresponds to a typical half-life of $\sim$ nanoseconds, $\Delta v_{1/2} = 2.8$ Hz). The $^1$H NMR spectrum has a single isotropic chemical shift for both of the protons within one $CH_2$ functional group (these are not diastereotropic), an indication of local flexibility quickly picked up by an edited $^{13}$C HSQC spectrum. Appreciation of the temporal dependence of free ligand exchange dynamics on NMR spectra can quickly inform medicinal chemists on local flexibility around bonds of newly synthesized molecules. This analysis, combined with potency data and matched molecular pair analysis, or a full comparison across a congeneric series, can provide critical insights into structure–activity relationships (Balazs et al., 2019).

Together with information regarding relative populations of conformations in solution, information about the magnitude of the rotational energy barrier between conformations, i.e., between one rotational isomer and another, is relevant information. The challenge has been to get quick, easy, and comprehensive yet accurate torsional profiles. Building a practicable and prospective visualization of conformational exchange dynamics around torsion angles into an NMR conformational analysis platform increases the potential to impact design by highlighting the potential energy penalty of restricting torsions. To evaluate the dynamics, incorporating REST-MD into the workflow met the goal of expanding the current free ligand conformational analysis platform to make use of kinetic parameters from NMR spectra, e.g., signal line widths, while keeping within practical time limit constraints for medicinal chemistry design cycles.

REST-MD predicts a comprehensive torsional profile in-silico for rotatable bonds represented in a 2D molecular structure while keeping computational speed and accuracy high. GPUs make REST-MD calculations feasible within drug discovery design cycle times. Ligand-based REST-MD simulates a ligand in explicit solvent at room temperature,

allowing for conformational effects often neglected due to computational expense. The ligand conformers are sampled according to their Boltzmann populations, and resultant reports visualize rotational torsion energies (Fig. 3). High-accuracy fragment-based calculations of rotational energy barriers (kcal mol$^{-1}$ TS2) are plotted as a function of bond torsion angle (solid lines). A superimposition of histograms counting the number of times the particular bond was observed at any particular angle is plotted onto the rotational energy barrier plot showing the torsion potential at each dihedral angle, summarizing the conformational space sampled during the REST-MD simulation. Such reports augment the $^1$H NMR spectral interpretation, providing quantitative energy minima and theoretical distributions of conformers.

### 2.2 NMR-measured rotational energy barrier

Methylation is a familiar and fundamental structural rigidification tool in a medicinal chemist's toolbox. In Fig. 2 methylation of the D ring demonstrates restricted bond rotation by the presence of rotameric signals in the $^1$H NMR spectrum. Such restricted bond rotations, on millisecond timescales, occur when barriers to rotation about a bond are high ($>\sim 15$ kcal mol$^{-1}$ under ambient conditions). In contrast, a structural analog without methyl groups on the D ring displays free bond rotation on the NMR timescale ($\sim$ nanoseconds). Typically such barriers to rotation at room temperature correspond to $\sim 5$ kcal mol$^{-1}$ (LaPlante et al., 2011a, b; Wipf et al., 2015). The NMR spectrum of the ensemble of rapidly exchanging conformations reflects the Boltzmann-weighted average of the chemical shifts, $J$ couplings, and interproton distances, with a single set of sharp peaks.

To locate the bond with the hindered rotation, chemical intuition is often sufficient. Using variable temperature NMR and/or exchange spectroscopy, the rotational energy barrier and the torsional rotation half-life of exchange can be determined. Figure 4 shows $^1$H NMR spectra as a function of eight different temperatures. The spectrum near room temperature has two equal rotameric populations undergoing slow exchange on the NMR timescale and highlighted in the figure. With increasing temperature the peaks coalesce and then begin to narrow. Increasing the temperature not only increases the rotation rate of the aromatic ring, but it also increases the rate of fluctuations in the pendant base and the $CH_2CF_3$ groups and between axial vs. equatorial methyl orientation in ring C. Overall, this drives a shift to higher ppm for the exchange-averaged signal with increasing temperature (Fig. 4).

In order to determine the barrier to rotation around the aromatic ring, it was important to collect data within a temperature range where exchange rates were dominated by the dynamics of interest in order to follow a simple two-state model for analysis. Therefore, three temperatures at 300, 305, and 310 K were chosen, and exchange spectroscopy was

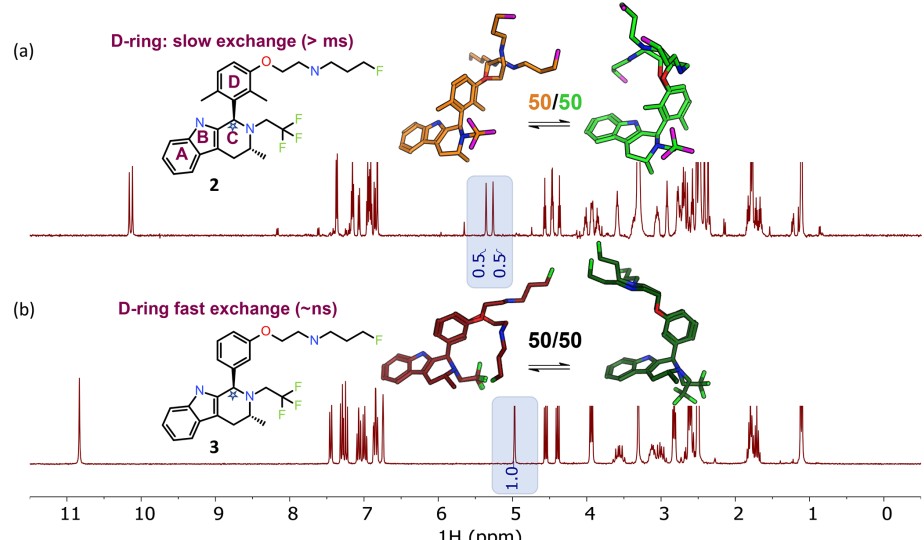

**Figure 2.** NMR spectra inherently capture kinetic information from conformational dynamics (rotational energy barriers) in the signal line widths. The benzylic CH (the starred CH in ring "C") is highlighted to exemplify the spectral changes between the dimethyl and des-methyl analogs. **(a)** The $^1$H NMR spectrum shows rotamers with equal populations undergoing slow exchange on the NMR timescale. Profiling of 2 gave pIC$_{50}$ 8.9 and logD 4.8, with 50 % bioactive conformation of the free ligand in DMSO-$d_6$ solution. **(b)** A spectrum with population-weighted conformational averaging due to low barriers to rotation around bonds and fast exchange on the NMR timescale. Profiling of 3 resulted in pIC$_{50}$ 8.8, logD 4.1, 50 % free ligand solution bioactive conformation. For this molecular matched pair we see similarities in the percent bioactive conformation in solution and the potency, regardless of the increased logD.

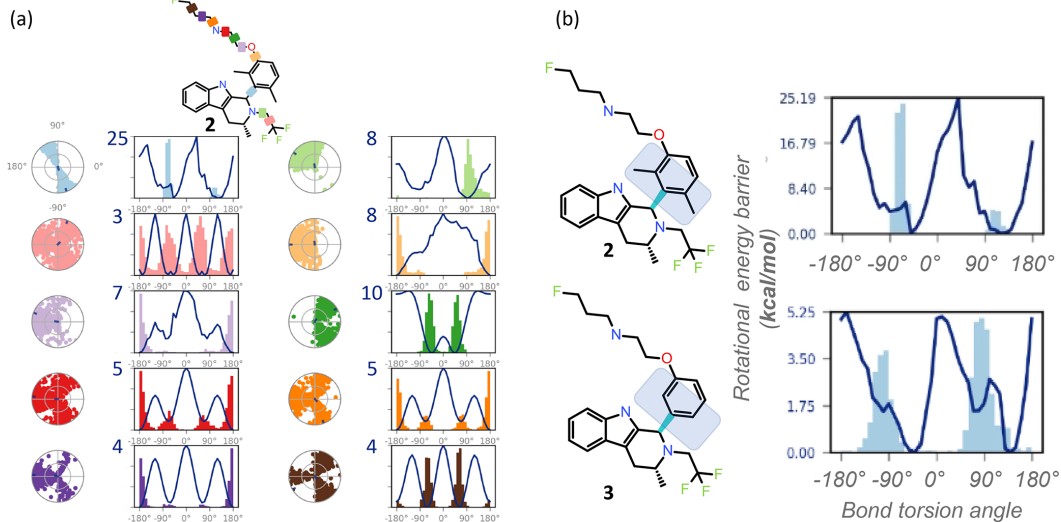

**Figure 3.** REST-MD visualizations, implemented in Maestro (Schrödinger, 2020b), complement atomic-resolution NMR interpretations of structural dynamics across all bonds. **(a)** Simulation interaction diagrams report the rotational energy barrier (kcal mol$^{-1}$) as a function of the bond rotation angle. The conformational space sampled during the simulation is reported either as a function of the simulation time (radial plots, from the center at the start and spiraling outward, with dark dots indicating initial and last sampled dihedral angles) or as histograms superimposed on the torsion energy profiles (kcal mol$^{-1}$ vs. bond angle across each color-coded bond in the molecule). Profiles are readily compared between molecular bonds that are pre-organized (light blue with a $y$-axis maximum in the plot at 25 kcal mol$^{-1}$, with a bimodal radial plot and two energy minima), and flexible (pink with 3 kcal mol$^{-1}$ maximum $y$-axis value, three energy minima, equally populated, and a randomly populated radial plot). **(b)** The barrier to rotation of the dimethyl is calculated to be, based on the lower of the two barriers, $\sim$ 20 kcal mol$^{-1}$. Whereas experimentally both energy minima are equally populated as seen by the 1 : 1 ratio by NMR, the sampling conditions of the rigid "blue" bond were insufficient in the simulation to equally populate both wells. The NMR data in such cases clearly inform on the calculated predictions. A separate REST-MD simulation for the des-methyl compound, 3, was a $<$ 6 kcal mol$^{-1}$ calculated rotational energy barrier for the same "blue" bond, consistent with sharp lines and ready conversions between the two conformers, with a broadened range of torsions, albeit still bimodal.

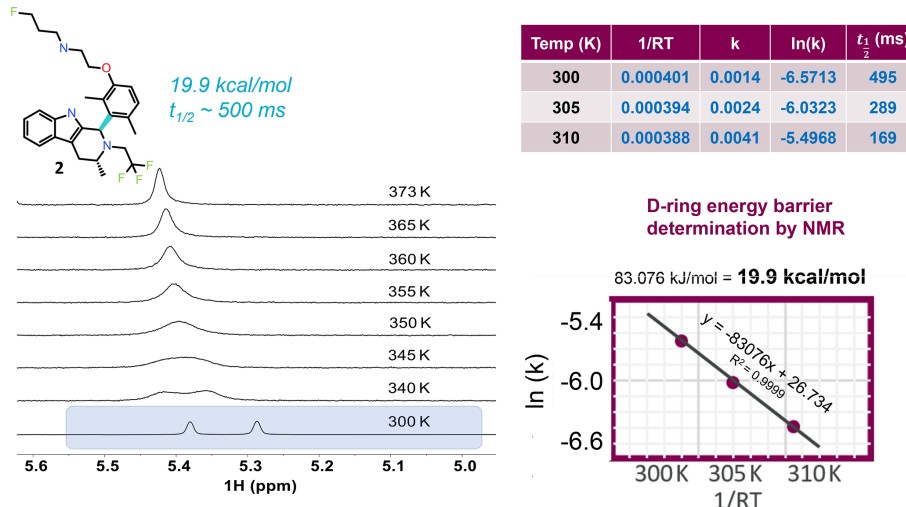

| Temp (K) | 1/RT | k | ln(k) | $t_{\frac{1}{2}}$ (ms) |
|---|---|---|---|---|
| **300** | 0.000401 | 0.0014 | -6.5713 | 495 |
| **305** | 0.000394 | 0.0024 | -6.0323 | 289 |
| **310** | 0.000388 | 0.0041 | -5.4968 | 169 |

**Figure 4.** VT NMR stacked plots for the dimethyl compound undergoing slow to fast exchange with increasing temperature. To measure the energy barrier to rotation of the D ring, three temperatures and a 1D selective EXSY at different mixing times were used to estimate the exchange rates and half-life. An Arrhenius plot gives the barrier to rotation at 19.9 kcal mol$^{-1}$, and the 300 K half-life is $\sim 0.5$ s.

performed with selective inversion on the peak near 5.3 ppm. At each of the three temperatures, eight mixing times (100, 200, 300, 400, 500, 700, 1000, and 2000 ms) were used to determine the first-order rate kinetics, with the fitted value for k given in tabular form in Fig. 4. The half-life was derived from ln(2)/k. The fitted plot of ln(k) vs. 1/RT is shown, revealing a value of 19.9 kcal mol$^{-1}$ for the barrier to rotation.

## 2.3 NMR informs calculations

REST-MD generates a large ensemble of ($\sim 1000$) conformations, in explicit solvent. REST-MD was run with Desmond (Schrödinger, 2020a) with the pendant base initially oriented either forward or backward relative to the tricyclic core for compound 1. The resultant calculated energy barrier of $\sim 20$ kcal mol$^{-1}$ (Fig. 3b) is in agreement with the NMR-determined value of 19.9 kcal mol$^{-1}$. The REST-MD visualization confirmed chemical intuition that the source of the rotameric species is the bond between the tricyclic core and the aromatic ring. Advantageously, the full torsional profile report from the REST-MD simulation can be run prospectively to rank design ideas, for instance to test a hypothesis around rigidification and the degree of bioactive pre-organization induced. The ability of REST-MD to evaluate torsion angles prior to synthesis can also flag chemists to check the NMR for site-specific dynamics information at the time of structural verification. Such information could alert the team to add a diagnostic selective-NOE measurement to the standard acquisition suite, to test a free ligand conformational hypothesis post synthesis, while the solution sample is in the spectrometer for structural identification.

Conversely, the NMR can supply experimental details inaccessible to the calculations, particularly helpful within a chemical series, as the lessons are generally translatable across the structural analogs. For instance, the $^1$H NMR spectrum of the dimethyl compound 2 showed a 1:1 ratio between the two exchanging conformations (Fig. 2), whereas REST-MD conformational sampling shows only one of the two minima populated (Fig. 5). The fragment-based energy calculation, shown as a solid line in the REST-MD torsional profiles, is consistently the same, even using a very short simulation time (e.g., picoseconds). The histograms vary based on initial conformation and number of replicates run. Starting with an initial conformation with the pendant base facing "forward" relative to the tricyclic core, the radial plot of torsion angle representation as a function of time starting at the center and spiraling outward only populates the $\sim +90°$ bond torsion angle during the 50 ns simulation that has a temperature range of 300–1263 K (12 replicates, 50 ns). Analogously, the histogram has one energy minimum populated, and the number of times the $\sim +90°$ torsion was observed is fairly narrowly distributed (Fig. 5, top). With the same initial conformation, increasing the temperature range to a high of 3302 K (20 replicates, 50 ns) showed some evidence of sampling of the opposite conformation in the radial plot (Fig. 5, middle). Starting the simulation with the 3D conformation switched to put the pendant base towards the back instead and running 20 replicates for a higher sampling temperature shows both minima were sampled (Fig. 5, bottom). Once this compound has been synthesized, it then becomes experimentally apparent from the 1H NMR spectrum at 300 K in DMSO-$d_6$ that both minima are equally populated (Fig. 2a). In this manner the experimental results can be fed back to

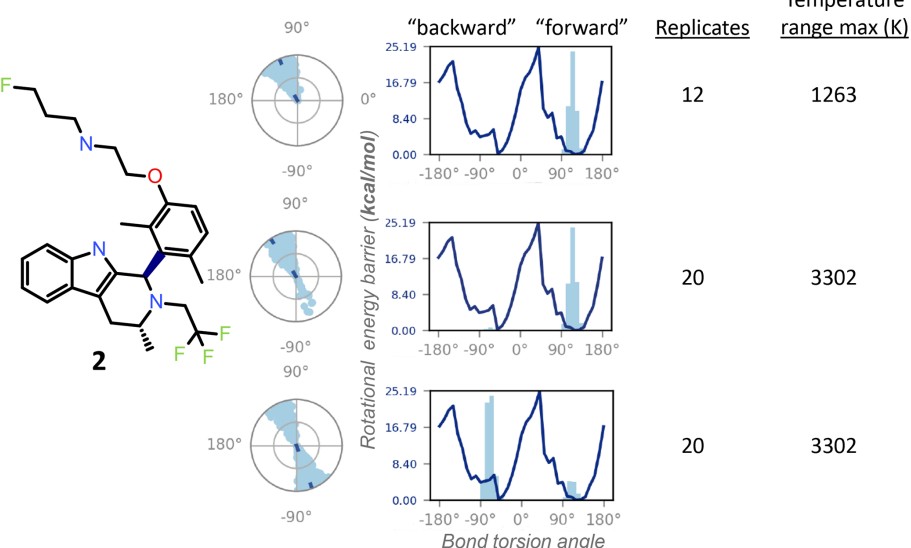

**Figure 5.** The REST-MD simulation of the dimethyl compound 2 predicts a preference to populate only one energy minimum, correlated with the initial dihedral angle starting condition in the simulation (dark dot in the center of the radial plots). Shown are results from calculations for conformations sampled as a function of bond rotation angle for the bond between the tricyclic core and the aromatic ring, bolded in the molecular structure. The three different simulation conditions are, from top to bottom: starting with the pendant base facing forward ($\sim +90°$) with a temperature range of 300–1263 K (12 replicates); the same initial conformation, with a temperature range of 300–3302 K (20 replicates); and the opposite initial conformation, with the pendant base facing backward, run with 20 replicates (300–3302 K). All REST-MD simulations ran for 50 ns.

the calculations to refine details and gauge areas of caution during interpretation.

## 2.4   REST-MD/NMR synergy in drug discovery

What REST-MD adds to the existing NMR platform is visualization of conformational dynamics by providing calculated rotational energy barriers across all bonds. This complements NMR spectral data to give insight into flexibility/rigidity at atomic resolution. Together, REST-MD and NMR conformational analysis allows us to utilize all the spectral information, thermodynamic and kinetic, gathered from $^1$H NMR spectra: chemical shifts, $J$ couplings, NOEs and line widths, to maximize characterization of free ligands in solution.

Without REST-MD, NMR alone can provide valuable information on the experimental conformational preference of the free ligand in solution. From the NMR alone, the relative populations of each conformer in solution can be deduced, and it can be determined whether the dominant conformer in solution is pre-organized into the bioactive conformation, which is of practical value for drug discovery. However, adding REST-MD provides an easy and practical way to visualize the magnitude of the energy penalty paid if the bioactive conformation is not highly populated in solution. This can help rationalize the cost-to-benefit ratio of design team effort invested in increasing the percentage of free ligand pre-organized into a bioactive conformation. As drug

discovery requires optimization of several parameters, knowing when binding has been optimized can shift design focus and resources towards improving physicochemical properties.

The added benefit of REST-MD is its ability to deliver prospective information regarding structural ideas of compounds not yet synthesized. Accurate predictions of free ligand solution conformational dynamics can help rank compounds to focus synthesis prioritization and flag supplemental experiments, such as selective NOEs on atom pairs to quickly ascertain expected conformations.

While the full torsional profile is powerful on its own, the REST-MD results also provide an extensively sampled conformational ensemble in explicit solvent that can be clustered and fed forward for use in NMR conformational analysis. Taking the selected conformer set forward for QM geometry refinements and calculations of NMR chemical shift and coupling constants provides the modeled parameter set used for further NAMFIS-based analysis. Using the conformer set generated by REST-MD is particularly helpful for higher molecular weight small molecules which begin to self-associate during low-mode MD conformational searches (LaBute, 2010) using a polarizable continuum model to emulate solvent. Low-mode MD provides a comprehensive search of the conformational ensemble but can therefore result in a set highly biased towards collapsed conformations.

## 3 Methods

### 3.1 $^1$H 1D and 2D ROESY solution NMR spectroscopy

$^1$H NMR spectra were recorded at 300 K on a 500 MHz NEO or a 600 MHz AVIII Bruker spectrometer with TCI cryoprobes. Solutions were made from 1 to 4 mg solid freshly dissolved in DMSO-$d_6$. Spectra were acquired with a 30° hard pulse, a 1 s delay, two dummy shots, and a signal averaged over 16 transients. A spectral width of $\sim 20$ ppm with 64 000 TS5 points was used. Spectral analysis was performed using an Advanced Chemistry Development, Inc. (ACD/Labs) Spectrus Processor (ACD/Labs, Version 2020.1.2). 2D ROESY (Schleucher et al., 1995; Thiele et al., 2009 TS6) was run with the Bruker standard pulse program roesyadjsphpr with ns 4, TD (1024, 256), and a 200 ms spin lock.

#### 3.1.1 Compound 1

$^1$H NMR (500 MHz, DMSO-d$_6$) Shift 10.54 (s, 1H TS7), 7.44 (d, $J = 7.7$ Hz, 1H), 7.23 (d, $J = 8.0$ Hz, 1H), 7.03 (td, $J = 7.9$, 1.2 Hz, 1H), 6.99 (t, $J = 8.2$ Hz, 1H), 6.97 (td, $J = 8.0$, 1.2 Hz, 1H), 6.89 (d, $J = 7.9$ Hz, 1H), 6.28 (br d, $J = 8.2$ Hz, 1H), 5.14 (s, 1H), 4.51 (dt, $J = 47.5$, 6.1 Hz, 2H), 4.00 (t, $J = 5.6$ Hz, 2H), 3.46 (dq, $J = 16.0$, 10.6 Hz, 1H), 3.37–3.33 (m, 1H), 2.99 (dq, $J = 16.0$, 10.0 Hz, 1H), 2.90 (t, $J = 5.7$ Hz, 2H), 2.78 (dd, $J = 16.0$, 4.5 Hz, 1H), 2.69 (t, $J = 6.9$ Hz, 2H), 2.62 (dd, $J = 16.0$, 7.7 Hz, 1H), 2.53–2.51 (m, 1H), 2.28 (s, 3H), 1.85–1.74 (m, 2H), 1.06 (d, $J = 6.7$ Hz, 3H) (https://doi.org/10.14272/GRTREOMLNXYRCK-CRICUBBOSA-N.1, Balazs, 2021a).

#### 3.1.2 Compound 2

$^1$H NMR (500 MHz, DMSO-d$_6$) Shift 10.18 (s, 0.5H, isomer1), 10.14 (s, 0.5H, isomer2), 7.39 (d, $J = 7.6$ Hz, 1H, isomer1+isomer2), 7.18 (t, $J = 7.2$ Hz, 1H, isomer1+isomer2), 7.09 (d, $J = 8.5$ Hz, 0.5H, isomer2), 6.99–6.95 (m, 1H, isomer1+isomer2), 6.95–6.90 (m, 1H, isomer1+isomer2), 6.88 (d, $J = 8.4$ Hz, 0.5H, isomer2), 6.84 (s, 1H, isomer1), 5.38 (s, 0.5H, isomer1), 5.29 (s, 0.5H, isomer2), 4.54 (dt, $J = 47.5$, 6.0 Hz, 1H, isomer1), 4.44 (dt, $J = 47.5$, 6.0 Hz, 1H, isomer2), 4.07–3.95 (m, 1H, isomer1), 3.95–3.85 (m, 1H, isomer2), 3.67–3.56 (m, 1H, isomer1+isomer2), 3.39 (s, 1H, isomer1+isomer2), 3.14-3.04 (m, 1H, isomer1+isomer2), 2.94 (br t, $J = 5.4$ Hz, 1H, isomer1), 2.80 (br t, $J = 5.6$ Hz, 1H, isomer2), 2.77 (br d, $J = 4.5$ Hz, 1H, isomer1+isomer2), 2.72 (t, $J = 7.0$ Hz, 1H, isomer1), 2.69 (br d, $J = 15.0$ Hz, 1H, isomer1+isomer2), 2.60 (t, $J = 6.9$ Hz, 1H, isomer1+isomer2), 2.44 (s, 1.5H, isomer2), 2.39 (s, 1.5H, isomer2), 1.83 (br dquin, $J = 26.2$, 6.6 Hz, 1H, isomer1), 1.82 (s, 1.5H, isomer2), 1.80 (s, 1.5H, isomer1), 1.72 (dquin, $J = 26.2$, 6.4 Hz, 1H, isomer2), 1.14 (d, $J = 6.5$ Hz, 3H, isomer1+isomer2)

(https://doi.org/10.14272/WCXZZJFDRDQAHJ-WXTAPIANSA-N.1, Balazs, 2021b).

#### 3.1.3 Compound 3

$^1$H NMR (600 MHz, DMSO-d$_6$) Shift 10.86 (s, 1H), 7.45 (d, $J = 7.8$ Hz, 1H), 7.31 (d, $J = 8.0$ Hz, 1H), 7.26 (t, $J = 7.8$ Hz, 1H), 7.08 (dd, $J = 8.0$, 7.5 Hz, 1H), 7.00 (dd, $J = 7.8$, 7.5 Hz, 1H), 6.86 (dd, $J = 8.2$, 2.4 Hz, 1H), 6.84 (d, $J = 7.8$ Hz, 1H), 6.75 (br d, $J = 2.3$ Hz, 1H), 4.98 (s, 1H), 4.48 (dt, $J = 47.5$, 6.6 Hz, 2H), 3.95 (t, $J = 5.6$ Hz, 2H), 3.57 (qd, $J = 13.0$, 9.3 Hz, 1H), 3.12 (dqd, $J = 11.0$, 6.8, 5.0 Hz, 1H), 3.01 (qd, $J = 18.0$, 9.3 Hz, 1H), 2.83 (t, $J = 5.7$ Hz, 2H), 2.64 (dd, $J = 15.8$, 5.0 Hz, 1H), 2.57 (dd, $J = 15.8$, 11.0 Hz, 1H), 2.62 (t, $J = 6.6$ Hz, 2H), 1.76 (dquin, $J = 26.1$, 6.6 Hz, 2H), 1.11 (d, $J = 6.8$ Hz, 3H) (https://doi.org/10.14272/UZTVFIJYDQIRSV-MZNJEOGPSA-N.1, Balazs, 2021c).

### 3.2 Exchange spectroscopy (EXSY)

$^1$H NMR spectra were recorded on a 500 MHz NEO at 300, 305, 310, 340, 345, 350, 355, 360, 365, and 373 K. For the 1D selective exchange spectroscopy at 300, 300, and 310 K, the mixing times used were 100, 200, 300, 400, 500, 700, 1000, and 2000 ms. The spectra were integrated with consistent integral ranges and by calibrating the integral of the inverted peak to 100 to consistently normalize the data (Hu and Krishnamurthy, 2006). An Excel spreadsheet was used to calculate the fractional intensity increase as a function of mixing time to fit exchange rate and half-life (Bovey, 1988; Li, et al., 2007).

### 3.3 REST-MD

Two different initial molecular conformations were run where the variation was placed in the relative position of the pendant base to the tricyclic ring: (i) "forward" or (ii) "backward", using the same atom numbering for all conformations sampled for the same compound, to simplify later steps in the workflow. Molecular protonation states at pH $7.0 \pm 0.0$ were used for the MD simulations. The force field builder in Maestro (Schrödinger, 2020b) was used to customize the OPLS3e force field for the system builder where a NaCl salt concentration of 0.15 M was used and the base was neutralized by addition of one Na+ ion during creation of the explicit water shell for solvation using the predefined SPC model and an orthorhombic box shape of $10 \text{ Å} \times 10 \text{ Å} \times 10 \text{ Å}$ using the "buffer" box size calculation method. Desmond (Schrödinger, 2020a) replica exchange with solute-tempering molecular dynamics was run with 12 replicas giving a temperature range of 300 K to typically $\sim 1300$ K, for a total of 50 ns for extensive sampling of conformational space during the simulation.

### 3.4    Simulation interaction diagram

The plots automatically generated in Maestro (Schrödinger, 2020b) provide solid lines tracing out the barrier to rotation in $kcal\,mol^{-1}$ as a function of the torsion angle. The histograms provide the resulting distribution of 1002 conformers under our routine sampling conditions. Radial plots show the evolution of the simulation time from the start, at the center, indicated by a dark dot and radiating outward until the final sampled conformer, also indicated by a dark dot.

### 3.5    Clustering of conformers

Ligands, without the solvent shell, were extracted from the REST-MD trajectories for both sets of initial conformers (forward and backward). To aid a quick visual inspection of the results, conformers was superimposed using the simplified molecular input lines entry system (SMILES) arbitrary target specification (SMARTS) method and the substructure SMILES string of c12c(C)c(CN)[nH]c1cccc2 to align the conformers relative to the rigid tricyclic ring. Conformers were clustered in Maestro (Schrödinger, 2020b) by atomic RMSD, discarding mirror-image conformers, selecting the option of one structure (nearest to centroid) per cluster, thus reducing the full set down to representative diverse conformers, typically 15–40.

### 3.6    QM calculations

Chemical Computing Group's (Molecular Operating Environment (MOE), 2019.01; MOE, 2021 TS8) conformational search graphical user interface was employed to generate input files for Gaussian 16 (Revision C.01, Frisch et al., 2016 TS9) after importing conformers into a molecular database and using the wash function to neutralize charged species not observed by NMR in DMSO-$d_6$ solutions. QM geometry refinement, chemical shift calculations and coupling constant calculations were carried out with the gauge including the atomic orbital (GIAO) density functional theory (DFT) method at the B3LYP/6-31G* level with PCM solvent modeling using a dielectric constant of 78.4. Geometry optimization keywords were set with opt=(tight,RecalcFC=5,MaxCycles=5000) and Int=SuperFineGrid.

### 3.7    Conformer distribution

MOE's Spectral Analysis graphical user interface was employed for least-squares fits of chemical shifts to determine conformer distributions; the option for couplings and NOEs was used selectively.

### 4    Conclusions

The REST-MD protocols described above provide rapid and prospective access to torsional energy barriers and conformational states for drug-like molecules. The REST-MD calculations accurately reproduce and visualize NMR dynamics which synergistically work with experimental conformational exchange dynamics obtained from 1D $^1$H NMR spectra. Integration of REST-MD into our NMR conformational analysis platform has enabled visualization of atomic-level information by all medicinal chemists and can be readily used to guide design hypotheses toward molecules with improved potency and/or physicochemical properties.

This new methodology has been applied across more than 10 early oncology projects in 2020, involving both small molecules and proteolysis targeting chimeras (PROTACs) drug leads, to answer questions around conformational preference (populations) and dynamics (rotational barriers).

In addition, NMR provides design teams with information on the presence of intramolecular hydrogen bonds (IMHBs) and the combined influences on properties such as potency, permeability and oral bioavailability. Diverse applications have enabled refinement of the approach and represent a step towards the goal of routine use for prospective design and determination of experimentally based conformation–activity relationships.

**Code availability.**   . TS10

**Data availability.**   . TS11 TS12

**Author contributions.**   AYSB and EC drove the earliest drafts of the manuscript, to which all the authors contributed. MJP developed and optimized computational workflows. DL and NLD determined the energy barrier to rotation by NMR. AYSB ran REST-MD simulations in Schrödinger and NAMFIS-based analyses in MOE. All the authors contributed valuable discussions to the preparation of this paper.

**Competing interests.**   All the authors are shareholders in AstraZeneca PLC.

**Special issue statement.**   This article is part of the special issue "Geoffrey Bodenhausen Festschrift". It is not associated with a conference. TS13

**Acknowledgements.**   The entire SERD team, including the analytical, computational, synthetic, and lead chemists, from the extended Oncology R&D group are thanked for chemical designs, syntheses, purifications, and compound profiling. We thank Jason Breed for the crystallography providing protein–ligand and bound conformations shown in Fig. 1. An Excel spreadsheet with detailed methodology for the 1D selective EXSY was kindly provided by

David Whittaker. We thank Jamie Scott and Michelle Lamb for valuable feedback on the manuscript.

**Review statement.** This paper was edited by Fabien Ferrage and reviewed by two anonymous referees.

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

## Remarks from the language copy-editor

CE1    "$k$"?

## Remarks from the typesetter

TS1    The composition of Figs. 2, 4, and the key figure has been adjusted to our standards.

TS2    Please note that units have been changed to exponential format. Please check all instances.

TS3    Please confirm.

TS4    Please confirm addition.

TS5    Please confirm.

TS6    Please confirm year.

TS7    Could you please define these letters (s, t, m, etc.). Should the numbers before "H" should be superscripted (as in [1]H above)?

TS8    Please confirm addition.

TS9    Please confirm addition.

TS10    Please provide a statement on how your underlying software code can be accessed. If the code is not publicly accessible, a detailed explanation of why this is the case is required. The best way to provide access to software code is by depositing it (as well as related metadata) in reliable public repositories, assigning digital object identifiers (DOIs), and properly citing code as individual contribution. Please indicate if different software codes are deposited in different repositories or if code from a third party was used. Additionally, please provide a reference list entry including creators, title, and date of last access. If no DOI is available, assets can be linked through persistent URLs to the software code itself (not to the repositories' home page). This is not seen as best practice and the persistence of the URL must be secured.

TS11    Please provide a statement on how your underlying research data can be accessed. If the data are not publicly accessible, a detailed explanation of why this is the case is required. The best way to provide access to data is by depositing them (as well as related metadata) in reliable public data repositories, assigning digital object identifiers (DOIs), and properly citing data sets as individual contributions. Please indicate if different data sets are deposited in different repositories or if data from a third party were used. If no DOI is available, assets can be linked through persistent URLs to the data set itself (not to the repositories' home page). This is not seen as best practice and the persistence of the URL must be secured.

TS12    Please mention Balazs, 2021a, b, c in this section.

TS13    Please confirm.

TS14    Please provide date of last access.

TS15    This reference is not cited in the text. Please check.

TS16    Please confirm the three data sets (taken from the assets tab).

TS17    Please check and confirm authors and title.

TS18    Please provide date of last access.

TS19    Please provide date of last access.

TS20    Please provide page range or article number.