# Peer review of "NMR free ligand conformations and atomic resolution dynamics"

_Magnetic Resonance, 2021_

## Author Response (AR1)

Instructions: The response to the Referees shall be structured in a clear and easy to follow sequence: (1) comments from Referees, (2) author's response, (3) author's changes in the manuscript. (Note on spectral DOI's is on p.7). In addition, please provide a marked-up manuscript version showing the changes made (using track changes in Word or latexdiff in LaTeX). This version should be combined with your response file so

5 that the Editor can clearly identify what changes have been made. (Note: track changes in Word begins on page 8).

(1) comments from Referees RC1:

General comments

- 10 Balazs et al. present an addition to their NAMFIS approach which now includes characterization of populations of alternate conformations and dynamics of the molecules under study. With the replica exchange approach, simulations times are virtually extended such that slow conformational exchanges can be accessed. The refined approach supports prospective ranking of design ideas in medicinal chemistry in a very efficient manner and sheds light onto alternative conformations and dynamics. Further, REST-MD indicates where potentially
- interesting alternate conformations and dynamics may occur which should be tested by NMR experiments, 15 thereby much simplifying and reducing the NMR analysis in a targeted manner.

**Specific comments**

30

Why were NMR experiments carried out in DMSO? This is somewhat puzzling since the MD simulations were carried out in explicit water with NaCl. I could well imagine that alternative conformations are populated

20 differently in DMSO than in aqueous buffer. Also, activity assays used to identify the bioactive conformation are carried out in aqueous buffer.

The MD protocol seems to yield very nice results for those torsion angles with an energy barrier < 10 kcal/mol. However, there are some questions about the rotation with the 25 kcal/mol barrier in the top right diagram in Figure 3b: Why aren't the two rotational energy barriers of identical height? The one at -170° seems to agree

25 with the experimental value, while the one at 10° seems significantly higher.

Figure 3b, top right diagram: the profile seems to imply lower energy for the +100° rotation angle, yet the populations are opposite.

The molecule presented is a very nice example of hindered rotation. However, if would be very instructive in order to assess the method, if more than just one example would be shown. But I guess that this is not possible due to trade secrets.

In order to reproduce such simulations, what would be your recommendation for the maximal temperature of the simulation? 3300 K looks a bit harsh, but seems to be required to compensate for short simulation times, even with replica exchange.

For prioritization of design ideas, relative populations of alternative conformations are important. The

- 35 populations in Figure 5 however don't seem to correlate with the actual populations observed by NMR. In your experience, which parameter of the simulation should be used to derive relative populations? Populations during MD (blue histograms), fragment based energy landscape (solid blue lines) or a combination of these parameters?
- 40 A very similar question comes up for how the dynamics in the MD correlate with the NMR experiment: Should the calculated energy barrier of rotation be used or how often an alternate conformation was visited in the MD?

Could references be provided that support the statements that NMR provides information on permeability and bioavailability etc.?

**RC2:**

In the present manuscript Chiarparin and coworkers describe the use of the REST-MD method (replica exchange with solute tempering in explicit water, introduced by the Friesner and Bern groups in 2005) as a means to investigate the conformational preference of small molecules in order to inform drug design. While I believe

- 50 that this manuscript presents solid science that would be publishable in principle (subject to minor revisions), I cannot see that it fits within the scope of Magnetic Resonance, because it does not present any advancements related to NMR. The manuscript describes a very brief case study that nicely illustrates the utility of REST-MD and the fact that NMR experiments can provide corrective results that augment the full picture during the drug design process. However, the use of NMR is limited to standard techniques to determine the rotamer
- 55 populations and the rotamer barrier of a single torsion in a pair of congeneric molecules, as a means to validate results from REST-MD. In part, the manuscript reads like a review of what has been (or could be) performed in other studies in terms of using NMR data to guide REST-MD calculations and other computational approaches. It is not at all clear what is actually new in this manuscript, except the specific results for this particular set of molecules.
- 60 I would suggest that the authors expand the scope of the manuscript by incorporating a larger set of NMR data, as the authors also suggest on p. 8, and clearly show how such data can be used to curate results from REST-MD calculations, or possibly introduce NMR-based constraints to guide calculations so as to truly make the calculations 'interpret' the NMR data. At present, the manuscript merely presents a brief (but nice) illustration of the use of REST-MD, followed by validation by NMR.
- 65 Minor points:

The REST-MD simulations involve explicit water, whereas the NMR studies used DMSO as solvent. Please comment on whether you expect deviations in rotamer populations due to the different solvents. Is it not possible to perform the NMR studies in water?

Fig. 2: Please highlight the benzylic CH group in the chemical structures to the left.

70 Fig. 2: Please define logD, as a service to readers outside of the medicinal chemistry field.

line 197-198: "...designing an increase in the percent bioactive conformation by restricting rotation". This sentence apparently confuses kinetics/dynamics with thermodynamics. Rotation is restricted by changing the barrier height, but this does not necessarily affect the relative populations of the two rotamer states, unless one of the two states is preferentially stabilized over the other, in which case it suffices to state just that, leaving the barrier (or restriction of rotation) out of the picture. There is similarly imprecise wording on lines 35-36.

line 207-210. I do not understand this sentence, please consider rephrasing. What is "low mode MD..."?

(2) author's response Reply on RC1:

75

Q: Why were NMR experiments carried out in DMSO? This is somewhat puzzling since the MD simulations were
 carried out in explicit water with NaCl. I could well imagine that alternative conformations are populated
 differently in DMSO than in aqueous buffer. Also, activity assays used to identify the bioactive conformation are
 carried out in aqueous buffer.

A: In our experience with small molecule drug candidates, in DMSO the conformer populations reproduce well those in buffer (pH 7); (for example: https://doi.org/10.1021/acs.jmedchem.9b00716). Therefore, we use DMSO

85 in routine application of this approach to ensure consistency across ligands, since solubility issues will often arise in pure water. This combination of water models with DMSO-derived data reflects the actual medicinal chemistry workflow.

Q: The MD protocol seems to yield very nice results for those torsion angles with an energy barrier < 10 kcal/mol. However, there are some questions about the rotation with the 25 kcal/mol barrier in the top right

90 diagram in Figure 3b: Why aren't the two rotational energy barriers of identical height? The one at -170° seems to agree with the experimental value, while the one at 10° seems significantly higher.

A: High rotational barriers will be subject to hysteresis and sampling effects and it is not possible to assign rigorous values to those barriers. The profiles are indicative of barrier height, but we rely on NMR data to rationalize these.

95 Q: Figure 3b, top right diagram: the profile seems to imply lower energy for the +100° rotation angle, yet the populations are opposite.

A: The minima lie at the same energy but sampling has been limited, as discussed further in Fig 5.

Q: The molecule presented is a very nice example of hindered rotation. However, if would be very instructive in order to assess the method, if more than just one example would be shown. But I guess that this is not

100 possible due to trade secrets.

> A: (Thanks!) Yes, the matched pair exemplar supports the complementarity of the computation and experiment for use as a platform to quide design, while keeping within the highest practicality for our current scope.

> Q: In order to reproduce such simulations, what would be your recommendation for the maximal temperature of the simulation? 3300 K looks a bit harsh, but seems to be required to compensate for short simulation times, even with replica exchange.

> A: In REST-MD the temperature is only a surrogate value, since the Hamiltonian is actually modified rather than the temperature. Other aroups use the terminology Hamiltonian replica exchange to distinguish this from standard temperature replica exchange. We show this very high surrogate temperature to demonstrate that sampling of high torsional barriers can be limited, even when the Hamiltonian is modified to mimic such a high

110 *temperature*.

105

Q: For prioritization of design ideas, relative populations of alternative conformations are important. The populations in Figure 5 however don't seem to correlate with the actual populations observed by NMR. In your experience, which parameter of the simulation should be used to derive relative populations? Populations during MD (blue histograms), fragment based energy landscape (solid blue lines) or a combination of these

115 parameters?

A: In a case where we see no issues with sampling of hindered rotations, as discussed in Fig 5, we use Boltzmann counting to assign simulation populations, after clustering using full-molecule RMSD. This would correspond to the circular histograms.

Q: A very similar question comes up for how the dynamics in the MD correlate with the NMR experiment: 120 Should the calculated energy barrier of rotation be used or how often an alternate conformation was visited in the MD?

A: In principle the MD trajectory should equilibrate to Boltzmann-weighted populations, consistent with the free energy of each conformational state. We use the computed torsion barriers to guide analysis of NMR data and look for consistency between barrier heights and NMR signals. As noted above, it is not guaranteed that an

125 unbiased MD trajectory will be able to explore all the conformers and cross all barriers to establish an equilibrium population. The most effective approach that we have found is to look at both experimental and computational data and to rely on models only when we have robust evidence that we are seeing trends consistent with experiment.

Q: Could references be provided that support the statements that NMR provides information on permeability and bioavailability etc.? 130

ie: p.12 lines 288-289: In addition, NMR provides design teams with information on the presence of intramolecular hydrogen bonds (IMHB), and the combined influences on properties such as potency, permeability and oral bioavailability

A: (1) Alex, A., et al., Med. Chem. Commun., 2011, 2, 669-674, https://doi.org/10.1039/C1MD00093D

135 (2) Over, B., et al., Nat Chem Biol., 2016, 12, 1065-1074, doi: 10.1038/nchembio.2203.

**Reply on RC2:**

**Q: Expansion of scope**

A: While NAMFIS and related methods provide conformer populations, the energy barriers and rotational
 dynamics are typically not quantified, despite being available from the NMR signal linewidth. This manuscript
 quantitatively compares for the first time, using a rotameric drug-like exemplar, measurements of both
 conformer populations and rotational dynamics with in silico calculated values. As discussed in section 2.4 on
 synergy of methods, our novel implementation of these tools are advancing the current state of the art in
 performing semi-automated structure and dynamics (the latter often underutilized) analyses of a free ligand
 in a drug discovery setting, including prospective molecular information, to guide drug design.

Q: introduction of NMR based restraints to simulations

A: While using NMR data to guide computational MD approaches is well established, it would not be prospective. Line 201 discusses the added benefit of delivering prospective information on molecules not yet synthesized. In a Design-Test cycle, for an exemplar of a chemical series of analogs experimental data is

150 generated. This is used to evaluate the limitations of the in silico data sufficiently well to confidently rank order design ideas and select which to progress on to real world, resource intensive, syntheses.

Q: Expect solvent effects on rotamers populations between DMSO and water?

A: We might expect population differences between apolar and polar solvent (https://pubs.acs.org/doi/pdf/10.1021/ja042890e). However, in our experience with small molecule drug

155 candidates, in DMSO the conformer populations reproduce well those in buffer (pH 7); (for example, https://doi.org/10.1021/acs.jmedchem.9b00716). We would not expect significantly different rotamer populations between DMSO and water in this molecule. An intramolecular energy barrier >20 kcal/mol should not depend on the solvent, population differences would not be expected in our experience.

Experimentally, compound (2) gives 1:1 rotamers when acquired in CDCl3 (below) and is insoluble in buffer, pH 160 7.

Q: Is it not possible to perform the NMR studies in water?

A: We believe DMSO is the most tractable option for this type of experimental platform. In consideration that the populations between DMSO and water are expected to be quite similar, and especially in consideration that

165 aqueous solubility is often quite low for small molecule candidate drugs, DMSO-d6 provides a consistent and pragmatic solution (https://doi.org/10.1021/acs.jmedchem.9b00716).

Request: Fig. 2: Please highlight the benzylic CH group in the chemical structures to the left.

Response: In the uploaded revised version: a star was added to the structures in Figure 2, and in line 91, "(the starred CH in ring 'C')" was added for clarity.

170 Request: Fig. 2: Please define logD, as a service to readers outside of the medicinal chemistry field.

Response: Added to the Figure 1 caption, line 49-50, "with inhibition ( $plC_{50}$ ) and the octanol/water partition coefficient (logD) values from a recent Oncology R&D project (Scott et al., 2016; Scott et al., 2019; Scott et al., 2020)" to clarify the first occurrences of the binding parameter, plC50, and the partition coefficient, logD.

Comment: line 197-198: "...designing an increase in the percent bioactive conformation by restricting rotation".
This sentence apparently confuses kinetics/dynamics with thermodynamics. Rotation is restricted by changing the barrier height, but this does not necessarily affect the relative populations of the two rotamer states, unless one of the two states is preferentially stabilized over the other, in which case it suffices to state just that, leaving the barrier (or restriction of rotation) out of the picture.

Response: Yes, thank you, this is about thermodynamics. To help clarify, the new line in the uploaded manuscript

180 is, "... of design team effort invested in increasing the percentage of free ligand pre-organized into a bioactive conformation." This should help distinguish that "design" refers to creation of a new molecule with increased population of bioactive conformation.

Comment: There is similarly imprecise wording on lines 35-36.

Response: updated sentence, added the words in bold (not bolded in the manuscript) "...the challenge is to conceive of ideas to modify the **structure to discover a new** molecule that favors the bioactive conformation."

Comment: line 207-210. I do not understand this sentence, please consider rephrasing. What is "low mode MD..."?

Response: line 207-210, after "low mode MD..." reference added in line: (LaBute, 2010) and to reference list:

Labute, P.,: LowModeMD-Implicit Low-Mode Velocity Filtering Applied to Conformational Search of Macrocycles and Protein Loops, J. Chem. Inf. Model., 50, 792-800, https://doi.org/10.1021/ci900508k, 2010.

(3) author's changes in the manuscript

195

200

205

- lines 35-36, added clarifying phrase to the sentence, "...the challenge is to conceive of ideas to modify the structure to discover a new molecule that favors the bioactive conformation."
- Figure 1 caption, line 49-50, "with inhibition (pIC50) and the octanol/water partition coefficient (logD) values from a recent Oncology R&D project (Scott et al., 2016; Scott et al., 2019; Scott et al., 2020)" to clarify the first occurrences of the binding parameter, pIC50, and the partition coefficient, logD.
  - Figure 2, a star was added to the structures, and line 91, "(the starred CH in ring 'C')" was added for clarity in the figure caption.
  - Line 198, to help clarify, the new line in the uploaded manuscript is, "...of design team effort invested in increasing the percentage of free ligand pre-organized into a bioactive conformation."
  - From Line 208, the following is the updated text, "Using the conformer set generated by REST-MD is
    particularly helpful for higher molecular weight small molecules which begin to self-associate during low
    mode MD conformational searches (LaBute, 2010) using a polarizable continuum model to emulate
    solvent. Low mode MD provides a comprehensive search of the conformational ensemble, but can
    therefore result in a set highly biased towards collapsed conformations."
  - Line 367, reference list, alphabetically added reference: Labute, P.,: LowModeMD-Implicit Low-Mode Velocity Filtering Applied to Conformational Search of Macrocycles and Protein Loops, J. Chem. Inf. Model., 50, 792-800, https://doi.org/10.1021/ci900508k, 2010.
- 210 **(4) Chemotion DOI's**, for 1D 1H NMR spectral database submissions (will work only after release of the embargo post-publication, https://www.chemotion.net/):

[revised manuscript text omitted]

---

## Author Response (AR2)

Dear Prof. Ferrage,

We are pleased with your decision to publish subject to corrections. Thank you for confirmation that the manuscript is within the scope of Magnetic Resonance.

5   We are also committed to Open Data. We are working with Chemotion repository for our $^1$H NMR spectra. The data has been accepted and the pre-publication embargo has been released. The live DOI links have been added under to the NMR multiplet string in the manuscript for each compound (p.10, lines 233, 244, and 252).

We have simplified Figure 1. There are fewer fonts and colors. Extraneous details to the illustration have been moved to the
10   caption or altogether. The four components of the design cycle have been labeled a-d, then explained in the figure caption.

[Figure]

**Figure 1. An illustrative Medicinal Chemistry design – make – test – analyze (DMTA) cycle for drug discovery; the example shown is taken from a recent Oncology R&D project (Scott et al., 2016; Scott et al., 2019; Scott et al., 2020). (a) Design: Medicinal Chemists design drug molecule ideas, using predicted values from computational models to prioritize which virtual molecules to**
15   **synthesize. (b) Make: Compounds are synthesized and NMR plays a key role for structural identification and analysis of compound purity. (c) Test: Compound profiling includes bioassays to quantify activity, such as target inhibition (pIC$_{50}$), and physico-chemical properties, such as the octanol/water partition coefficient (log$D$). X-ray structure of the bound ligand-protein complex and NMR free ligand conformations are measured, including relative population of the bioactive bound conformation. (Balazs et al., 2019). (d) Analyze: compound free and bound structures are analyzed against measured properties to rationalize**
20   **structure activity and property relationships to derive new hypotheses for improved designs in step (a) of the cycle. Typical discovery projects comprise ~1000 cycles from hit to drug candidate.**

In Figure 5, p.8 line 187, we added, "(dark dot in the center of the radial plots)." In addition, in Figure 3, when the radial plots are first described, we've added, for further clarity, to "The conformational space sampled during the simulation is reported either as a function of the simulation time (radial plots, from the center at the start and spiraling outward" the following text: p.5 lines 117-118, ", with dark dots indicating initial and last sampled dihedral angles) ..." Similarly, to the methods section, page 11, line 269 after, "Radial plots show the evolution of the simulation time from the start, at the center", we've included for clarity, "indicated by a dark dot, and radiating outward until the final sampled conformer, also indicated by a dark dot."

Section 3.1, the title, "NMR spectroscopy" has been made more specific: $^1$H 1D and 2D ROESY solution NMR Spectroscopy. In this section, p.9 line 225, Schleucher et al., 1995; Thiele et al., 2008, the two relevant EASY ROESY references were added in line and page 16, in the full references were added alphabetically.

Line 410, Schleucher, J., Quant, J., Glaser, S. J., Griesinger, C.: A Theorem Relating Cross-Relaxation and Hartmann-Hahn Transfer in Multiple-Pulse Sequences. Optimal Suppression of TOCSY Transfer in ROESY, J. Magn. Reson. A, 112, 144-151, https://doi.org/10.1006/jmra.1995.1025, 1995.

Line 435, Thiele, C., Petzold, K. and Schleucher, J.: EASY ROESY: Reliable Cross-Peak Integration in Adiabatic Symmetrized ROESY, Chem Eur J., 15, 585-588, https://doi.org/10.1002/chem.200802027, 2009.

The last changes were to expand acronyms at first use:

p.10, l.250, the acronym (EXSY) was added after exchange spectroscopy, as EXSY is used on p.13 line 314 in the acknowledgements sections.

p. 11 l.274, SMARTS was expanded prior to the use of the acronym.

p.11 l.280, GIAO DFT

p.12 l.287 GUI was replaced by graphical user interface in the text.

p.12 l.297 proteolysis targeting chimeras (PROTACs)

We thank you for this excellent opportunity to participate with a celebratory contribution to Geoffrey Bodenhausen's Festschrift.

On behalf of all the authors,

Amber Balazs